



# TIMED Doppler Interferometer Measurements of Neutral Winds at the Mesosphere and Lower Thermosphere and Comparison to Meteor Radar Winds

Arthur Gauthier[1,2,3], Claudia Borries[3], Alexander Kozlovsky[4], Diego Janches[5], Peter Brown[6,7], Denis Vida[6], Christoph Jacobi[8], Damian Murphy[9], Masaki Tsutsumi[10,11], Njål Gulbrandsen[12], Satonori Nozawa[13], Mark Lester[14], Johan Kero[15], Nicholas Mitchell[16,17], Tracy Moffat-Griffin[16], and Gunter Stober[1,2]

[1]Institue of Applied Physics, University of Bern, Bern, Switzerland
[2]Oeschger Center for Climate Change Research, University of Bern, Bern, Switzerland
[3]Institute for Solar-Terrestrial Physics, German Aerospace Center (DLR), Neustrelitz, Germany
[4]Sodankylä Geophysical Observatory, University of Oulu, Finland
[5]ITM Physics Laboratory, NASA Goddard Space Flight Center, Greenbelt, MD, USA
[6]Dept. of Physics and Astronomy, University of Western Ontario, London, Ontario, Canada
[7]Western Institute for Earth and Space Exploration, University of Western Ontario, London, Ontario, Canada
[8]Institute for Meteorology, Leipzig University, Leipzig, Germany
[9]Australian Antarctic Division, Kingston, Tasmania, Australia
[10]National Institute of Polar Research, Tachikawa, Japan
[11]The Graduate University for Advanced Studies (SOKENDAI), Tokyo, Japan
[12]Tromsø Geophysical Observatory, UiT - The Arctic University of Norway, Tromsø, Norway
[13]Institute for Space-Earth Environmental Research, Nagoya University, Japan
[14]University of Leicester, Leicester, UK
[15]Swedish Institute of Space Physics (IRF), Kiruna, Sweden
[16]British Antarctic Survey, Cambridge, CB3 0ET, UK
[17]Department of Electronic and Electrical Engineering, University of Bath, Bath, UK

**Correspondence:** Arthur Gauthier (arthur.gauthier@dlr.de) and Gunter Stober (gunter.stober@unibe.ch)

**Abstract.** The mesosphere and lower thermosphere (MLT) is a highly variable region that forms the transition region between middle and upper atmosphere. The variability of this region is driven by atmospheric waves transporting energy and momentum from the lower and middle atmosphere to MLT altitudes. These waves cover a wide range of temporal (minutes to days) and spatial (kilometers to planetary) scales. The upward propagation of atmospheric gravity waves and tides is one of the key processes at all latitudes that alters the state of the ionosphere-thermosphere system and their vertical propagation depends crucially on the background mean winds. The TIMED Doppler Interferometer (TIDI) onboard the Thermosphere-Ionosphere-Mesosphere-Energetics and Dynamics (TIMED) satellite observes neutral winds at the MLT using airglow emissions. We establish a TIDI mean wind climatology, compare our results with existing climatologies derived from local meteor radar observations, and discuss similarities and differences depending on local time and geographical latitude.



## 1 Introduction

Wind measurements with global coverage at the mesosphere/lower thermosphere (MLT) are still sparse. These winds are characterized by atmospheric waves of various spatial and temporal scales and provide a substantial source of variability at the lower boundary of the upper atmosphere. Hence, they play a crucial role in the vertical coupling between the middle and upper atmosphere affecting space weather (Liu, 2016). Atmospheric solar tides contribute substantially to this variability and have

wave periods corresponding to an integer fraction of a day. Forced by the absorption of solar radiation due to water vapor in the troposphere, ozone in the stratosphere, or by molecular or atomic absorption at thermospheric heights (Lindzen and Chapman, 1969; Lindzen, 1979), atmospheric tides gain large amplitudes in the MLT, depositing some energy and momentum into the mean flow (Becker, 2017). Zonal mean MLT winds reflect a characteristic climatology for both horizontal components. Zonal winds show a hemispheric summer wind reversal from westward to eastward wind around 90 km altitude and moderate/weak

eastward winds during the hemispheric winter season. The meridional wind exhibits a southward prevailing wind during the northern hemispheric summer and a corresponding opposite behavior in the southern hemisphere. These wind systems are closely related to the residual circulation and associated gravity wave drag (Lindzen, 1981; Smith, 2012; Becker, 2012).

Wind measurements providing global coverage including both polar regions independent of the Yaw-Cycle maneuvers are performed by the TIMED spacecraft. Due to the spacecraft orbit, TIDI observes all local times for each location between

the Yaw-Cycles every 60 days. Thus, TIDI vector winds are affected by migrating and non-migrating tides (Wu et al., 2006; Oberheide et al., 2005) as well as planetary wave oscillations. This poses additional challenges in determining mean winds, which are already considered less reliable due to larger uncertainties in the zero calibration of the instrument (Niciejewski et al., 2006).

This study compares TIDI winds to ground-based meteor radar observations covering northern and southern polar and mid-

latitudes between 2003 and 2020. Based on these comparisons, we develop additional statistical quality control filters, which are generalized, to obtain globally harmonized TIDI winds for composite days. These composite days are compiled from 60-day averages for all longitude and latitude bins to mitigate potential offsets due to varying measurement statistics between night and day and issues related to the zero line calibration due to night-day transition orbits. Further validation of TIDI winds and comparison with other systems is necessary. Meteor radar winds have proven to provide a reliable benchmark for cross-

comparisons with General Circulation Models (GCMs)(Pokhotelov et al., 2018; Stober et al., 2021b). Furthermore, meteor radar (MR) winds were used for the validation of meteorological analysis such as the Navy Global Environment Model - High Altitude (NAVGEM-HA) (McCormack et al., 2017; Stober et al., 2020; Liu et al., 2022; van Caspel et al., 2023) or for data assimilation. MR winds are processed using the same averaging as applied to TIDI zonal and meridional winds to ensure the best possible spatial and temporal overlap and to mitigate the intrinsic differences in the sampling.

The goal of this study is to establish a TIDI mean wind climatology with processed TIDI wind data and compare it with climatologies derived from local meteor radar observations. The manuscript is structured as follows: in Section 1, the analysis of TIDI winds and its comparison to meteor radar winds for different stations are presented. A seasonal comparison in zonal and meridional winds for meteor radar stations at different latitudes is introduced in Section 2. Figures 5 and 6 in this section





highlight the correlation between TIDI and MR winds for both zonal and meridional components. In Section 3, the study of TIDI wind-latitude cross-sections highlights the latitude-altitude relation for zonal winds: an indication of the summer hemispheric wind reversal is shown in this section. Finally, in this section, Figure 8, TIDI global coverage demonstrates on a global map the presence of diurnal and semi-diurnal tides due to the solar heating of the atmosphere.

## 2 Instrumentation and wind data

### 2.1 TIDI observations

TIDI is a Fabry-Perot interferometer developed and built by the University of Michigan and designed to investigate the dynamics of the Earth's mesosphere and lower thermosphere-ionosphere (MLTI). Its observational mode ranges from 70 to 120 km during the daytime and from 80 to 103 km at night. TIDI observes neutral winds at different altitudes using several airglow emission lines of $O2(0-0)$ at 765.07 nm, 763.78 nm, and 764.00 nm, and daytime O 557.7 nm at higher altitudes (Killeen et al., 2006).

TIDI comprises four telescopes that are orthogonally orientated (two at 45° forward on either side of the satellite's velocity vector and two at 45° rearward of the satellite) so that wind vectors on both sides of the satellite track can be observed. These 4 views enable the construction of the vector winds as a function of altitude along two parallel tracks (cold side and warm side of the satellite). Due to the orthogonality of the two telescopes on the same side of the spacecraft, the same locations are observed with a delay of a few minutes when the satellite moves along its track (Wu and Ridley, 2023).

TIMED orbits Earth about 15 times every day. Its precession rate of 12 minutes per day and the combination of ascending and descending orbits determine that the TIDI sampling track covers 24 hours in local time every 60 days. In addition, the orbital parameters were chosen so that TIDI reaches the initial latitude and LST coordinates after exactly one calendar year again. Niciejewski et al. (2006).

TIDI has continuously taken data since 2002 except for a data gap in early 2003. A decrease in the throughput of the instrument and an increase in cross-talk between the telescopes was observed after launch, which was determined to be caused by ice on the detector housing. Also, it was observed that TIDI had a light leak which led to uncertainties in the inverted wind data. A satellite maneuver was performed in January 2003 which enabled TIDI to reach nominal performance (Killeen et al., 2006).

There exist two different ways of retrieving the neutral winds: one is used by the University of Michigan, and the other is used by the National Center for Atmospheric Research (NCAR). While the University of Michigan linearly interpolates the samplings of each telescope to an evenly spaced track angle grid and calculates the wind vectors at these grid points, NCAR pairs sampling locations from one telescope with the nearest neighbors from the other (Killeen et al., 2006). The results of both these schemes do not present a lot of differences. In this study, we use data from the University of Michigan http://tidi.engin.umich.edu. The vertical altitude resolution of the instrument is 2.5 km. Each acquisition cycle takes between 100 and 200 seconds to complete which gives an along-track resolution of around 750 km. Warm and cold side neutral wind vectors give respectively the zonal (eastward) and meridional (northward) wind components through calculation from the line-





of-sight (LOS) winds measured in the two directions on the same side of the spacecraft.

## 2.2 Meteor radar (MR) winds

MRs are standard instruments to measure MLT winds. Thus, our comparison includes only MRs that have a long observation
80   record at mid- and polar latitudes. MR winds are often analyzed by fitting a least square function projecting all measured
line-of-sight velocities onto mean horizontal winds (Hocking et al., 2001; Holdsworth et al., 2004). In this study, zonal and
meridional winds are obtained by applying modified retrievals including non-linear error propagation (Gudadze et al., 2019;
Hindley et al., 2022), a Tikhonov regularization for the vertical wind (Stober et al., 2022), and World Geodetic Coordinate
System 84 (WGS84) projections to correct the line-of-sight velocities for the Earth's ellipsoidal geometry (Stober et al., 2021a).
Figure 1 presents a geographic projection of all MRs that are used for the comparison to the TIDI zonal and meridional winds.

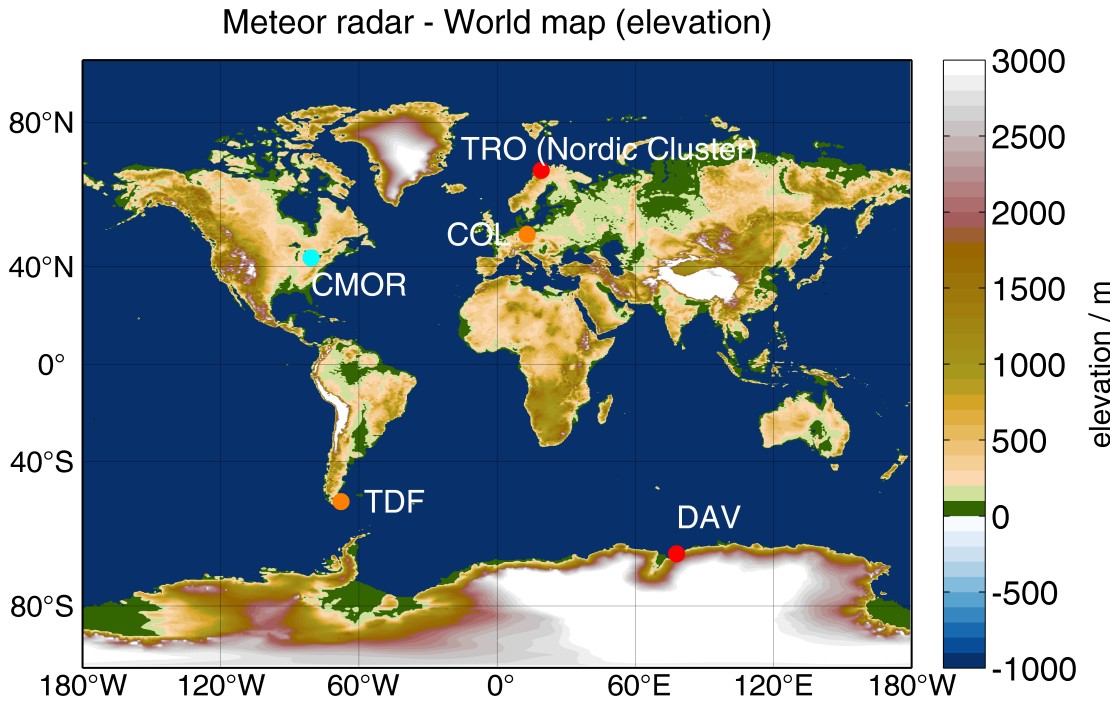

**Figure 1.** Geographic map of the MRs included in the validation of the TIDI winds. The plot was generated from etopo2 using the m_map
package (Amante and Eakins, 2009).





|  | latitude | longitude | references |
|---|---|---|---|
| DAV (Davis) | 68.58°S | 77.97°E | Holdsworth et al. (2008) |
| TDF (Tierra del Fuego) | 53.79°S | 67.75°W | Fritts et al. (2010) |
| CMOR | 43.26°N | 80.77°W | Webster et al. (2004) |
|  |  |  | Brown et al. (2010) |
| TRO (Nordic) | 69.6°N | 19.2°E | Hall and Tsutsumi (2013) |
|  |  |  | Stober et al. (2021a) |
| COL (Collm) | 51.31°N | 13.0°E | Jacobi et al. (2007) |

**Table 1.** Geographic location MRs used for the comparison to TIDI winds.

All of these stations have in common that the MRs have been in operation for more than a decade. The Nordic Meteor Radar Cluster consists of 5 MRs located in Fennoscandia at Tromsø (TRO) since 2003, Alta (ALT) since 2014, Kiruna (KIR) since 1999, Sodankylä (SOD) since the end of 2008, and Svalbard (SVA). However, the mainland radars in Norway, Sweden, and Finland are located in such a narrow geographic region that we treat them as one system in our analysis and refer always to TRO. We compiled a merged data set from all four mainland systems from 2003 to 2023. At mid-latitudes, we leveraged observations from Collm (COL) and the Canadian Meteor Orbit Radar (CMOR). In the Southern hemisphere, the MRs located at Tierra del Fuego (TDF) at Rio Grande Argentina, and Davis (DAV) are included in the comparison. References for each system can be found in Table 1.

MR winds are computed using a vertical resolution of 2 km between 70-120 km. The temporal resolution is one hour. However, to match the temporal averaging of the satellite, the hourly MR winds are averaged ±30 days for every hour of the day around a center day to obtain composites with a temporal resolution of one hour. These composites are compared to the TIDI zonal and meridional winds and used to optimize the filtering and averaging of the TIDI vector winds.

## 3 TIDI winds analysis and MR wind comparison

### 3.1 TIDI - MR wind comparison and quality control criteria

TIDI vector winds are provided over a certain geographic location and time with a vertical resolution of 2.5 km between 70-120 km. Firstly, we remove all zonal and meridional wind measurements that exceed 120 m/s. In the next step, we bin the data in longitude, latitude, and time. We prepare a spatial grid with 30° longitude and 10° latitude bins. Furthermore, we average all measurements that fall within each longitude and latitude bin for each hour using all orbits within ±30 days to ensure the sampling of all local times for each spatial and temporal bin. Additionally, we calculate the geophysical variance and number of accepted observations with each bin. Both quantities turn out to be useful for further quality measures in our comparison.




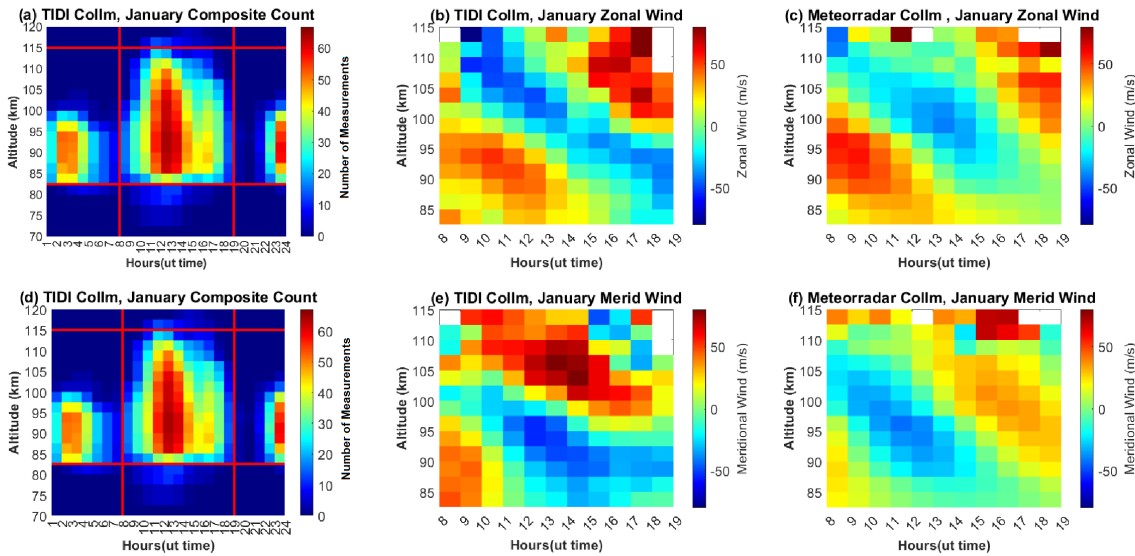

**Figure 2.** Comparison of TIDI and MR winds. left panels: Number of successful TIDI measurements of the time of the day and altitude with the longitude and latitude bin where the MR is located. central panels: TIDI zonal and meridional daily composite winds, respectively. right panels: Zonal and meridional daily composite MR winds for Collm.

For the TIDI and MR wind comparison, we use the longitude/latitude bin of the TIDI winds where the MR is located. If several MRs are within one bin, we performed only one comparison: in the case of the Nordic Meteor Radar Cluster we choose the one at Tromsoe for our comparison. We computed hourly composites for both instruments. Figure 2 shows the comparison of zonal and meridional winds for the Collm MR and TIDI for January. The left upper panel represents the number of TIDI measurements for each altitude and time based on the 60-day composites. The horizontal and vertical red lines label the area with the highest number of measurements. The central and right upper panels show the resultant altitude-time zonal winds for TIDI and the MR at Collm. Zonal winds exhibit a prominent semidiurnal tide and a remarkable agreement between both data sets, which is typical for the winter months at the mid-latitudes (Stober et al., 2021b).

Furthermore, we set a threshold requiring the number of TIDI observations per hour and altitude to be more than 30. Stricter filtering, demanding even higher measurement statistics of TIDI, increases the correlation but at the cost of a decreased number of available zonal and meridional winds. Therefore, we only accepted TIDI zonal and meridional winds with measurement statistics of 30 or more TIDI observations for each time, altitude, and latitude-longitude bin. During the hemispheric winter, we found a strong correlation for TIDI and meteor radar measurements of the zonal wind (R=0.87) and a moderate correlation for the meridional wind measurements (R=0.67). Furthermore, TIDI winds appeared to be more reliable during local daytime and exhibit a decreased agreement during the night hours. Figure 3 visualizes the same comparison for the Collm MR during





|  | zonal (summer) | merid(summer) | zonal(winter) | merid(winter) |
|---|---|---|---|---|
| DAV (Davis) | 0.62 | 0.78 | 0.69 | 0.49 |
| TDF (Tierra del Fuego) | 0.70 |  | 0.67 | 0.37 |
| CMO (CMOR) | 0.44 | 0.82 | 0.52 |  |
| TRO (Nordic) | 0.71 | 0.61 | 0.52 | 0.69 |
| COL (Collm) | 0.67 | 0.56 | 0.87 | 0.67 |

**Table 2.** Correlations between TIDI and all meteor radars for both zonal and meridional winds in winter and summer

summer for both wind components. Although the measurement statics appear to be increased during some times of the day between 82-110 km compared to the winter months, the overall correlation for both wind components is lower. During the summer months, we find correlations of R=0.67 and R=0.56 for the zonal and meridional winds, respectively. A summary of all comparisons for all radar locations for the summer and winter months is given in Table 2. The gaps in the table indicate locations for which less than 30 observations were made.

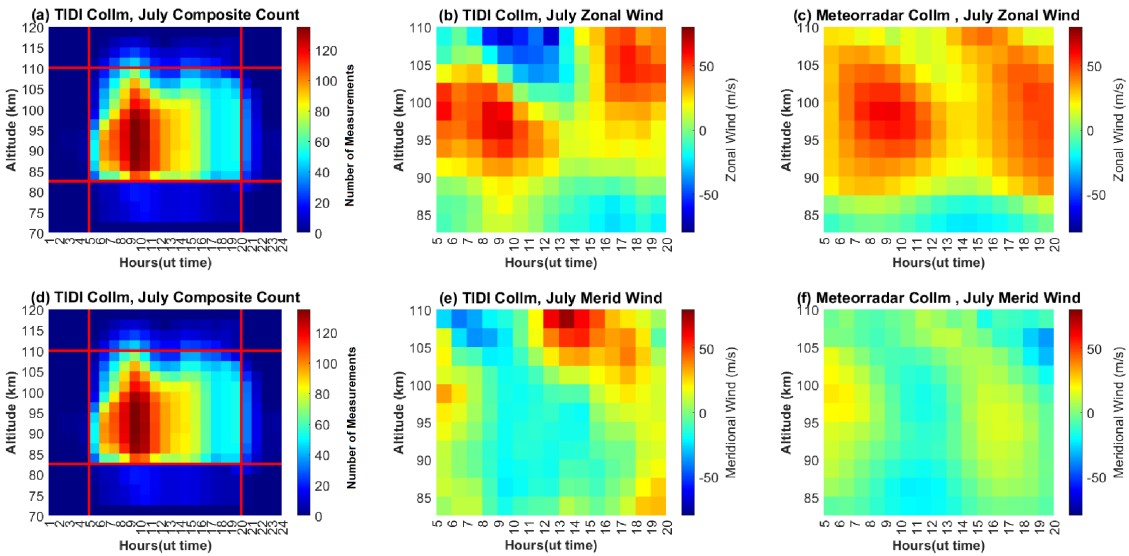

**Figure 3.** The same as Figure 2, but for July.


## 3.2 Seasonal TIDI - MR winds comparison

A seasonal comparison of the zonal-mean zonal and meridional wind can be found in Figure 4 for all three mid-latitude sites. These zonal-mean zonal and meridional winds are constructed from composite days of accepted quality wind compilations





**Figure 4.** Seasonal comparison of zonal and meridional winds for the mid-latitude station Collm, CMOR and TdF and the zonal mean zonal and meridional TIDI winds for corresponding latitude.



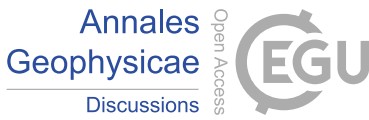

**Figure 5.** Correlation and density plots comparing the zonal mean zonal and meridional IDI winds to the MR climatologies within the latitude bin for all mid-latitude stations COL, CMOR and TdF (RioGrande). The red line denotes the ideal correlation. The black lines are the slopes to the data for both linear fits considering always the other variable as independent. The $h$ describes the t-test result. h=1 indicates the null-hypothesis was rejected, h=0 reflects the acceptance of the null-hypothesis.

for each longitude and latitude bin. We mitigated the different data coverage between daytime and nighttime by building a

composite day averaging all observations for all longitudes, latitudes, and time-altitude bins within 60-days. These composite





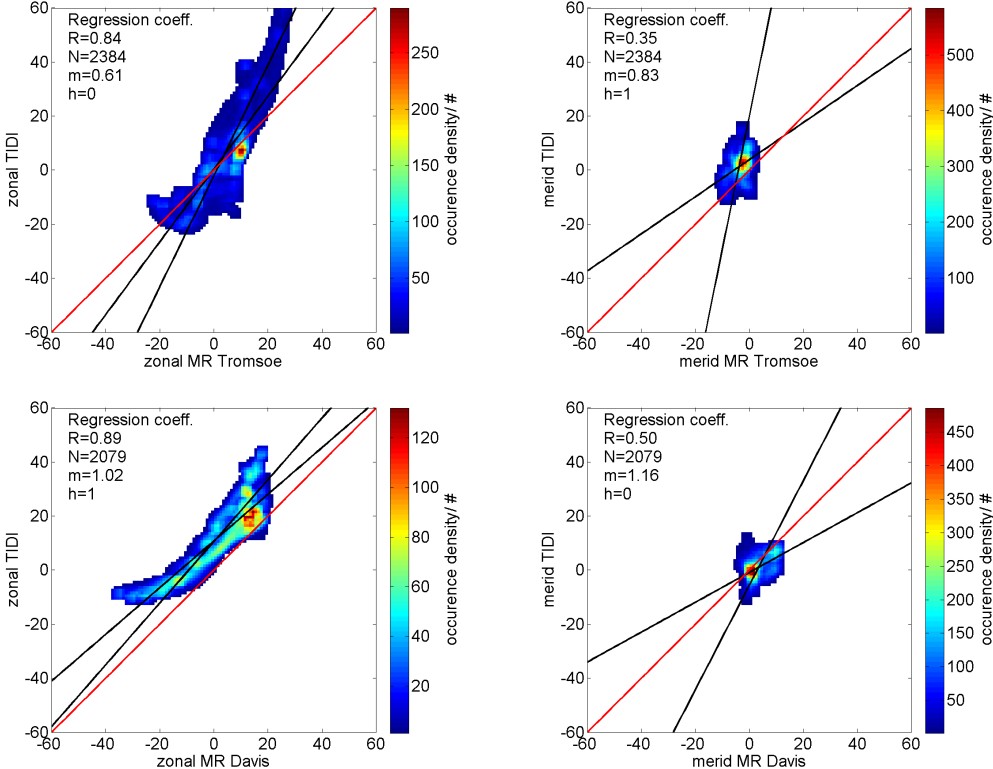

**Figure 6.** The same as Figure 5, but for polar latitudes at TRO and DAV.

days are then decomposed by applying a tidal fit including mean zonal and meridional winds, and the diurnal, semidiurnal, and terdiurnal migrating and non-migrating tidal components. However, due to the long averaging that is required, we model non-migrating tides as noise. This means that the non-migrating tides are being fitted but ignored in the rest of the study. In particular, at mid- and polar-latitudes the geophysical variability due to various tidal components and planetary waves within 135 60-days seemed to be problematic for inferring reliable non-migrating tidal components. We applied the same global tidal fit as in Baumgarten and Stober (2019);

For the construction of a composite year from the multiple years for the seasonal comparison, every composite year is associated with the year it is belonging to. For each day the mean value between different years is taken.

$$u, v = u_0, v_0 + \sum_{s=-3}^{3} \sum_{n=1}^{3} A_{sn} \sin(s \cdot \lambda - 2\pi \cdot t/T_n) + B_{sn} \cos(s \cdot \lambda - 2\pi \cdot t/T_n) \tag{1}$$

where $T_n$ corresponds to periods of 24, 12, and 8 hours, $s$ denotes the zonal wave number, and $\lambda$ is the geographic longitude. $A_{sn}$ and $B_{sn}$ are the Fourier coefficients for each wavenumber and period, $u_0$ and $v_0$ describe the zonal-mean zonal and meridional wind. This fit significantly improves the robustness of the derived zonal and meridional mean winds, as day-night differences in the TIDI data quality can be modeled as tidal modes and, thus, reduce the biases for both mean wind components.





However, the estimated migrating and non-migrating tidal modes appear to be less reliable due to the intermittency of the tidal

amplitudes and phases as well as the planetary wave activity within 60-days and longitude dependent offsets in the TIDI data quality and measurement statistics. The derived zonal-mean zonal and meridional winds are then compared to the MR climatologies applying the same averaging window of 60-days. The seasonal comparison shows that TIDI zonal winds capture most of the seasonal characteristics at mid- and polar latitudes compared to the MR climatology (Figure 4) below 95 km. The summer zonal wind reversal from westward to eastward winds is well-reproduced concerning the reversal altitude and

magnitudes as well as the weak eastward winds during the winter season up to about 90-94 km altitude. TIDI winds also reflect the gradual height change of the summer wind reversal altitude during the summer months for the northern mid-latitudes. Also, the asymmetry between spring and fall transition is visible at lower latitudes, with eastward winds being present at 85km in September for TIDI. At this height, almost zero winds are observed in April. The spring transition from the winter to the summer seems to exhibit some difference. MR zonal winds for COL and CMOR show a westward wind over the entire MLT,

whereas TIDI shows a weakening of the eastward winds. Winter zonal winds differ; while MR winds show a slow decrease of westerlies with height, TIDI winds are westward above 90 km. In the southern hemisphere at TdF, the spring and fall transition seem to show less deviation compared to the TdF winds. This remaining differences seem to be a result of the 60-days averaging required for TIDI, which results in a reduced sensitivity to such more rapid changes in the mean winds that last only for a few weeks.

Meridional winds show in general a less good agreement compared to the MR observations for all stations. However, even TIDI meridional winds catch the general seasonal wind morphology below 95 km altitude. TIDI winds show a significant southward wind during the summer months and weak northward during the hemispheric winter season, which resembles the MR wind pattern qualitatively. However, it remains unclear why TIDI meridional winds indicate southward winds during the winter months in the northern hemisphere. Such a circulation is absent for COL and CMOR.

We quantified the agreement between the MR observations and TIDI winds by computing correlation and performing a paired t-test to identify potential biases assuming that ideally, both distributions should have the same mean. Figures 5 and Figure 6 show the correlations for the climatological TIDI and MR zonal-mean zonal and meridional winds at mid- and polar- latitudes, respectively. TIDI zonal winds show correlation coefficients between R=0.67-0.8. The highest correlation is found for TIDI winds at the polar latitude at TRO, whereas the lowest correlation was obtained for DAV. The other MRs at COL, CMOR, and

TDF show correlations between R=0.7 and R=0.76. TIDI meridional winds exhibit only a very low correlation compared to the MR winds. At polar latitudes, beyond 55° on both hemispheres, these seasonal correlations are limited to hemispheric summer months when we apply our quality criteria. The t-test accepts the null-hypothesis ($h = 0$) for mid- and low-latitude stations for the zonal wind and occasionally for the meridional component. However, a rejection of the null-hypothesis indicates a substantial bias of the mean wind larger than 2 m/s.

## 3.3  TIDI winds latitude cross section

Given the reasonable agreement of the TIDI zonal winds and the MR measurements, we calculated latitude cross-sections of the zonal mean zonal and meridional winds for a typical northern hemispheric winter/summer condition in January and





July, which are shown in Figure 7. The latitude-height cross-section visualizes the hemispheric asymmetry of the zonal and meridional winds. TIDI captures differences in the strength of the eastward zonal jets between the northern and southern

hemispheres. Furthermore, TIDI winds indicate a weak eastward wind up to about 90 km altitude during hemispheric winter. These winds are in good agreement with the MRs and the meteorological analysis NAVGEM-HA (McCormack et al., 2017; Stober et al., 2020; Liu et al., 2022). Our analysis also supports the presence of a hemispheric asymmetry found in the meteor radar data with stronger eastward winds during northern summer conditions compared to southern polar latitudes (Stober et al., 2021b). The wind systems during hemispheric winter exhibit eastward wind from polar latitudes down to about 20° latitude,

and a similar wind pattern is observable for the northern hemisphere winter.

During the hemispheric summer months, TIDI meridional winds at mid- and polar latitudes also show a characteristic vertical wind structure indicating equatorward winds during the hemispheric summer below 90 km, as expected from the residual circulation, and poleward inflow of air at higher altitudes. Equatorial latitudes and the corresponding winter hemisphere reflect only weak meridional winds.

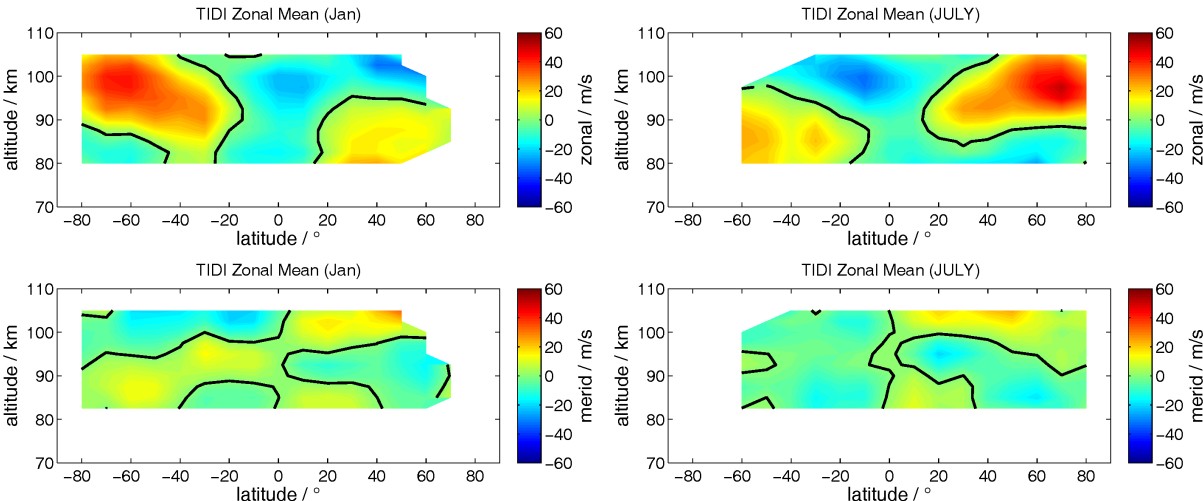

**Figure 7.** Latitudinal cross-section of zonal mean zonal winds from TIDI for January (left panel) and July (right panel).

Finally, Figure 8 shows global projections of TIDI winds using 60-day averages to construct a composite day. The six panels represent snapshots of the zonal wind component based on 4-hour means during January. The presence of negative zonal wind (westward wind) at low latitude regions and positive zonal wind (eastward wind) at middle and high latitudes is expected. It is possible to see in 4h steps how the structure of the winds moves over time and to distinguish the diurnal westward propagating tide due to the solar heating of the atmosphere in the southern hemisphere and the SW2 migrating tide at mid- and polar

latitudes on the northern hemisphere.



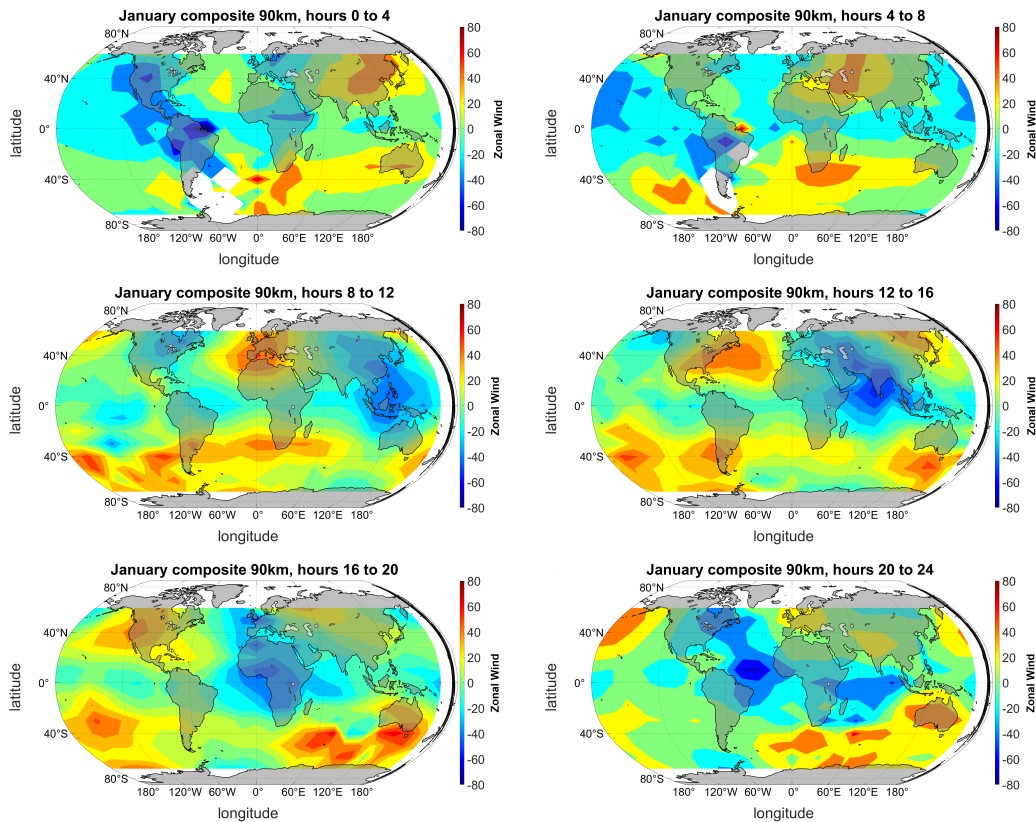

**Figure 8.** These pictures represent zonal winds for the composite month of January. These pictures are compiled based on 60-day averages using the quality criteria derived in this paper.

## 4 Discussion

Neutral winds are important, as they have a strong influence on the vertical propagation of atmospheric waves in the MLT region. The comparison between ground-based meteor radar measurements and satellite observations at these heights can prove particularly useful for improving data analysis and processing for both measurement systems. Our study shows very good correlations between TIDI and meteor radar zonal winds depending on local time and geographical latitude. TIDI seasonal zonal winds are well reproduced compared to the MR winds at mid-latitudes and during hemispheric summer conditions at polar latitudes. However, TIDI winds sometimes indicate a bias compared to the MR, which is likely associated with TIDI calibration as well as geophysical effects due to non-stationary atmospheric waves.

Our comparison of TIDI zonal and meridional winds and MRs at mid- and polar latitudes shows overall reasonable agreement of the general wind morphology for all altitudes below 95-98 km. TIDI captures the hemispheric summer zonal wind reversals and shows eastward winds during hemispheric winter up to 95 km. Only during the spring and fall transitions larger discrepancies are evident relative to the MR. The latitude-height cross sections of the derived climatologies also confirm the results



of previous studies based on multiple MR or meteorological analyses such as NAVGEM-HA (McCormack et al., 2017; Stober et al., 2020; Hindley et al., 2022).

We tested the null hypothesis with a paired t-test for the seasonal zonal and meridional winds to investigate the influence of the zero line calibration. This test compares whether the MR winds and TIDI observations have the same mean to within a 2 m/s threshold. This constitutes our null hypothesis. This threshold is to take into account the fact that the observation volumes are still quite different. The null hypothesis was occasionally rejected at polar and mid-latitudes on both hemispheres for zonal and meridional winds, although the correlation coefficients reached 0.7-0.8 during the hemispheric summer months at TRO

or DAV. The offset was on the order of up to 5-15 m/s. Overall, TIDI zonal winds seem to be more reliable at low and mid-latitudes, whereas the polar latitude sites at TRO and DAV exhibit larger deviations from the one correlation line. Meridional winds reflect a substantial deviation compared to the MRs at all latitudes, which is reflected in much lower correlations between R=0.27 to R=0.5.

Furthermore, the inferred latitude-altitude cross sections for typical hemispheric summer/winter conditions underline that the

zonal mean zonal TIDI winds are eastward up to 95-100 km during the hemispheric winter and also captures the latitudinal dependence of the summer wind reversal height, which increases with latitude. TIDI winds also reveal the hemispheric asymmetry of the magnitude of the summer eastward jets (Stober et al., 2021b).

TIDI meridional winds also reflect the general morphology of the seasonal wind characteristics compared to MRs up to an altitude of 95 km. Above this height, the winds seem to indicate substantial differences and appear to be biased when considering

the MR winds as a reference. Based on the latitude-height cross-section, meridional winds exhibit an equatorward flow during the hemispheric summer months below 90 km, which is consistent with the expected behavior due to the residual circulation and mesospheric cooling due to the upwelling within the polar cap.

In this study, we focussed on a detailed zonal-mean zonal and meridional wind TIDI wind analysis mitigating potential technical issues due to the zero-line calibration. The global tidal fit based on 60-day composite days showed clear signs of additional

factors that caused substantial biases. Atmospheric tides as well as planetary waves have a significant variability within the required 60-day window. This variability is relevant for the amplitudes, but also for the phase stability of these waves, which causes information mixing between migrating and non-migrating tides and other atmospheric waves. In particular, the Quasi-two-Day Wave (Q2DW) can reach large amplitudes of 40-60 m/s for a few days and has often a zonal wavenumber of 3 and, thus, affects all longitudes and several latitudes in one hemisphere (Stober et al., 2024). Additionally, other transient and rapidly

evolving events such as Sudden Stratospheric Warmings or the spring and fall transitions (Stober et al., 2020; Matthias et al., 2021) are problematic in constructing the composite day and cause larger uncertainties. Gravity waves are also contributing to the overall noise, but are less critical due to their more local nature.

## 5 Conclusion

Analyzing data from TIDI and MRs at different latitudes, this study focuses on the wind comparison of zonal-mean zonal

and meridional winds and on dependence of local time and latitude. For that purpose, TIDI data is binned in a spatial grid





in latitude, longitude, and time. The TIDI winds are compared to the MR winds for co-located grid cells. This comparison is performed for 5 MRs for both zonal and meridional components in the summer and winter months. We derived a quality control filter based on the number of measurements per longitude-latitude and time altitude to ensure sufficient statistics for TIDI taking into account the unequal daytime and nighttime measurement distribution. The correlation calculation between

TIDI measurements and different MR stations shows a reasonable agreement for zonal winds and reveals larger deviations for the meridional component.

The TIDI-MR seasonal comparison for mid-latitude stations at COL, CMOR, and TdF between 50° N and S indicates that TIDI captures most seasonal characteristics for zonal winds. This is also true for zonal winds at polar latitudes during hemispheric summer. In particular, the summer zonal wind reversal from westward to eastward winds is observable. For meridional

winds, the seasonal comparison is still reasonable, at least for TIDI winds up to 95km. However, meridional winds generally indicate a less good agreement with the MR wind climatologies than the zonal component. TIDI winds appear to be most reliable between the equatorial to mid-latitudes and exhibit biases up to 5-16 m/s at polar latitudes for the meridional wind component in hemispheric summer conditions. This work also highlights the altitude-latitude dependence for TIDI winds. For the zonal-mean zonal and meridional component, TIDI winds show the expected summer hemispheric wind reversal and the

latitudinal dependence on the reversal altitude. This is visible even at higher latitudes. Our study supports the presence of a hemispheric asymmetry between the summer and winter hemispheres, with stronger eastward winds at higher latitudes for the summer hemisphere compared to eastward winds at latitudes for the winter hemisphere. For the meridional component and below 90 km, the presence of equatorward winds during hemispheric summer is revealed through the vertical wind structure at higher latitudes. Finally, this study demonstrates that zonal mean TIDI climatologies at mid-latitudes and during hemispheric

summer conditions are well reproducing the mean circulation to be compared with the MRs. Thus, TIDI winds might be useful as a lower boundary for general circulation models such as TIE-GCM.

Our analysis also outlines a potential to estimate the zero line calibration for lower-level TIDI data products by introducing statistical bias corrections that are altitude, latitude, longitude, and time-dependent. This can either be optimized by a generalized Tikhonov matrix or through machine learning approaches assuming MR winds or other climatologies as ground truth.

*Author contributions.* AG developed all source codes to optimize the global data binning of the TIDI winds and performed the data analysis. The global tidal fitting routine was developed by GS and Kathrin Baumgarten and adopted for this study. All Authors contributed to the editing of the manuscript.

*Competing interests.* GS, CB, are handling editor at AnnGeo, CJ is chief editor at AnnGeo.



*Acknowledgements.* Gunter Stober, Arthur Gauthier are members of the Oeschger Center for Climate Change Research (OCCR). The Es-
range meteor radar operation, maintenance, and data collection were provided by the Esrange Space Center of the Swedish Space Corporation.

This research has been supported by the STFC (grant no. ST/W00089X/1 to Mark Lester).

Njål Gulbrandsen acknowledges the support of the Leibniz Institute of Atmospheric Physics (IAP), Kühlungsborn, Germany, for their
contributions to the upgrade of the TRO meteor radar.

Operation of the Davis meteor radar was supported by Australian Antarctic Science projects 4445 and 4637. Support for Diego Janches
as well as SAAMER-OS' operation are provided by NASA's Planetary Science Division Research Program, through ISFM work packet
Exospheres, Ionospheres, Magnetospheres Modeling at Goddard Space Flight Center and NASA Engineering Safety Center (NESC) as-
sessment TI-17-01204. This work was supported in part by the NASA Meteoroid Environment Office under cooperative agreement no.
80NSSC21M0073.

This research has been supported by the Schweizerischer Nationalfonds zur Förderung der Wissenschaftlichen Forschung (grant no.
200021-200517/1), the Science and Technology Facilities Council (grant no. ST/W00089X/1), the Japan Society for the Promotion of Sci-
ence (grant nos. 21H04516, 21H04518, 21H01144, and 20K20940), the Australian Antarctic Division (AAS grant nos. 2668, 4025, 4445 and
4637), NASA Engineering and Safety Center (grant no. TI-17-01204), NASA Engineering and Safety Center (grant no. 80NSSC21M0073),
the US NSF (grant no. AGS-1651464), the Deutsche Forschungsgemeinschaft (grant no. JA 836/47-1), and the International Space Sci-
ence Institute (ISSI) in Bern (through ISSI International Team project 23-580 – Meteors and Phenomena at the Boundary between Earth's
Atmosphere and Outer Space).





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
