# Peer review of "TIMED Doppler Interferometer Measurements of Neutral Winds at the Mesosphere and Lower Thermosphere and Comparison to Meteor Radar Winds"

_Annales Geophysicae, 2024_

## Referee Comment (RC2)

[referee-annotated manuscript omitted]

---

## Author Response (AR1)

**Reply to reviewer 1**

**General Reply:**

We thank the reviewer for the assessment of our manuscript and the confirmation of the conclusions presented in the paper. The main focus of the presented manuscript is the compare mean wind climatologies between TIDI and local meteor radars. Furthermore, we intended to apply a statistical quality control based on the available information from previous publications such as Niciejewski et al., 2006, Wu et al., 2008 and Wu et al., 2023. The statistical method described in the manuscript has the great advantage that no information about the satellite in orbit geometry is required. Furthermore, we aimed to assess with en cross-validation the standard TIDI observations as they were released.

We also appreciate the recommended literature. There are six publications listed below, where three are already cited and two are outside the main narrative of the presented comparisons. In particular, TIDI comparisons to ICON/MIGTHY are limited to 10S up to 40 N, whereas in this paper we compare TIDI winds at mid and polar latitudes on both hemispheres.

Finally, our results do not contradict previous findings presented in one of the recommended paper from Dhadly et al., 2021.

**General Comment:**

This study aims to establish a mean wind climatology using TIDI data, compare the results with climatologies derived from five meteor radar stations located around the globe, and examine the seasonal and altitudinal similarities and differences between them. The TIDI winds database, as the most extensive archive of global MLT neutral winds, offers a valuable opportunity to compare climatological variations in MLT winds with those measured by ground-based instruments.

However, I have significant concerns regarding this study. Below are my specific comments and issues that must be addressed before this manuscript can be considered for acceptance.

There are two major issues with TIDI wind measurements that the author has not addressed or discussed in the manuscript.

**Comment:**

First, TIDI winds have known bias issues. This bias was recently discovered in 2021 when TIDI winds were compared with MIGHTI winds in the MLT region at their conjunctions [e.g., Wu et al., 2023; Dhadly et al., 2021]. The TIDI bias is complex, as it varies with the yaw cycle -- of which there are nearly six each year. Additionally, the bias is telescope-dependent, with four independent TIDI telescopes [Wu et al., 2023]. This annual and cyclical bias has likely contaminated many of the results presented in this study (also likely the reason for strong summer differences (Figure 3) compared to winter (Figure 2)). These bias issues must be addressed before comparing TIDI winds with MR winds. One potential solution is to exclude data from periods when TIDI winds are biased. See details in Dhadly et al. [2021] where TIDI winds are compared with ICON/MIGHTI winds.

**Reply:**

TIDI has been in orbit for many years and the data set has been available since then but did not gain the same attention compared to the SABER instrument onboard the same satellite. Our goal was to use the existing data and to derive a climatological comparison to ground based instruments at high and polar latitudes. A climatological cross-comparison was not performed so far. The implemented statistical quality control was designed to optimize the data filtering without the need for detailed orbital characteristics of the spacecraft as this information is not available for the standard wind data product. The statistical method addresses the issues raised by the reviewer due to the median and on a second stage due to the adaptive spectral fitting to suppress and mitigate remaining issues related to the cold and warm telescopes. The statistical filter was designed leveraging the information provided in Wu and Riddley, 2023 as cited in the manuscript. The method presented by Dhadly et al., 2021 is certainly helpful in that regard. However, we must not that section 2.2 of Dhadly et al. 2021 essentially cites the same papers than we summarized in technical description. We will add the findings of Dhadly et al., 2021 in our discussion of our bias results. We noticed the jumps in the wind speed as well and mitigated the issue by our adaptive spectral filter approach given in eq. 1.

Line 144-146 "However, the estimated migrating and non-migrating tidal modes appear to be less reliable due to the intermittency of the tidal amplitudes and phases as well as the planetary wave activity within 60-days and longitude dependent offsets in the TIDI data quality and measurement statistics."

We will add to this sentence explicitly the warm and cold side bias of TIDI.

**Comment:**

The second concern relates to TIDI measurement errors around the terminators. The available TIDI data includes measurements at all local times, including around the terminators. Wind measurements from optical sensors (such as TIDI and MIGHTI) near terminators—where there is a gradient in emissions - tend to have significant uncertainties. TIDI suffers from twilight contamination and thus is not recommended when the solar-beta angle is high [e.g., Niciejewski et al., 2006; Dhadly et al., 2018]. The manuscript lacks any discussion of uncertainty analysis or how uncertainties might explain discrepancies between TIDI and MR winds. Additionally, no quality control was applied to filter out high-uncertainty winds near the terminators, which would ensure that only high-quality TIDI data is used in the comparisons. One solution would be to exclude data where the solar beta angle is high, though this would result in 15-20 day gaps in the TIDI dataset during each yaw cycle [e.g., Wu et al., 2023; Dhadly et al., 2024; Niciejewski et al., 2006]. These gaps would complicate the 60-day averaging for tidal fitting, though interpolation could potentially resolve this issue.

**Reply:**

This issue is also mitigated by our statistical approach. As shown in Figure 2 the terminator times are well picked up by our statistical criterion. During these times a minimum of measurements is available and, thus, our threshold filtering suppresses a contamination. The adaptive spectral filter is also used for the zonal mean zonal and meridional wind fit and can deal with such data gaps. We implemented a version that is dedicated to handle irregular sampled time series. We performed tests with composite day by converting all longitudes in local times as well as by doing to opposite. However, we did find clearly jumps in the wind that we also addressed to warm and cold biases between the telescopes that prevented us from performing a tidal comparison to the local meteor radar observations.

**Comment:**

Both of these issues have likely compromised the results and must be addressed before any meaningful comparison with MR winds can be made. If the biased periods are excluded and the terminator data filtered out, I believe the comparison between TIDI and MR winds would show significant improvement.

**Reply:**

We disagree to the reviewer conclusion, which is speculative. We mitigate both issues through fitting and statistical quality control. Given the high correlation of the zonal winds for the mid- and high latitude stations and slopes close to 1 (m-parameter) for Collm, Tromso and Davis a systematic contamination is unlikely. As also noted in the manuscript more strict criteria for filtering might improve the correlation for a few locations, but results in even larger data gaps. Furthermore, all other comparisons were done to ICON MIGHTY and for latitudes around the equation between 10 S and 40 N.

**Comment:**

Referencing: In my opinion, the manuscript lacks adequate referencing of previous literature relevant to this study, particularly those that have utilized TIDI winds in the MLT region for similar analyses or comparisons with space-based sensors. For instance, Wu et al. [2008] studied TIDI winds and discussed zonal mean zonal winds. Since the TIDI bias likely began around the 2012-2014 period, the findings from Wu et al. [2008] remain valid and should be included in the context of this study, but they are currently missing from the manuscript.

**Reply:**

Wu et al., 2008 is referenced. We will expand our discussion of the zonal mean zonal and meridional winds.

**Comment:**

Lines 66-72: The author likely derived this information from another study, as neither Michigan nor NCAR contributed to this work. Please cite the source of this information (for example personal communication and/or a study discussing it)

**Reply:**

This information was derived from the article:

TIMED Doppler Interferometer: Overview and recent results - Killeen - 2006 - Journal of Geophysical Research: Space Physics - Wiley Online Library

I will cite this source in the manuscript.

**Comment:**

Lines 74-76: This sentence is confusing and possibly incorrect. Zonal and meridional winds are derived on both sides of the spacecraft (cold and warm sides), meaning that vector winds are available on both sides of the satellite velocity vector.

**Reply:**

We will rephrase this sentence to avoid misunderstanding.

**Comment:**

Section 3: Including a figure (similar to the first column of Figure 2) showing the annual distribution of MR and TIDI winds would be helpful.

**Reply:**

MR winds show no significant change in the altitude coverage concerning this study. We have statistical uncertainties for each altitude-time bin based on non-linear error propagation. The statistical uncertainties show a diurnal pattern due to diurnal variation of the meteor flux.

**Comment:**

Figure 2: A figure showing the error distribution would be extremely helpful for explaining the discrepancies between TIDI and MR winds.

**Reply:**

We will add a figure including the MR errors in the appendix.

**Comment:**

Lines 112-113: While the agreement appears good qualitatively, the winds seem quite different quantitatively. I suggest rephrasing this to say that the overall behavior shows similarities rather than claiming "remarkable agreement."

**Reply:**

**Agreed.**

**Comment:**

Lines 119-120: I don't see a clear change in the agreement between TIDI and MR winds with local time; they appear similar during both day and night hours.

**Reply:**

The contour plots are just showing daytime zonal and meridional winds according to our statistical filter.

**Comment:**

Table 2: I assume that January and July are being referred to as winter and summer, respectively. If so, please specify this in the caption. Additionally, July may not be a suitable month for TIDI comparisons because, by the end of the month, TIDI might be observing near the terminator, which could result in more uncertain wind measurements. Please verify the local time of TIDI measurements in July at the locations of the MR stations.

**Reply:**

We will revise the caption.

**Minor Comments:**

**Comment:**

Line 121: What does "measurement statistics" refer to? Please clarify in the text.

**Reply:**

Measurement statistics refers to the number of available wind measurements released in the data base for a specific time-altitude-longitude bin. We will rephrase to avoid ambiguity.

**Comment:**

Figure 4: This figure is confusing. Does it use data from all TIDI longitudes? If not, why is it limited to only three stations? If it does include all longitudes, then this is not a true apples-to-apples comparison, and it needs to be reconsidered.

**Reply:**

The zonal mean zonal and meridional wind climatologies are compiled using all longitudes that passed through the quality control and all local times. Meteor radar wind climatologies have been proven to be representative for the zonal mean wind structure (https://doi.org/10.5194/acp-20-11979-2020). Due to the issues with the terminator and the warm and cold bias a comparison to the local climatology seemed to even more misleading.

**Comment:**

Figure 4: Please include units (such as km and m/s) in brackets. The current format is confusing.

**Reply:**

The Journal accepts the format.

**Comment:**

Figure 5: There is a typo in the caption: "IDI winds" should be corrected to "TIDI winds."

**Reply:**

Done.

**Comment:**

Figure 5: Please clarify the linear fitting used in this figure. It is unusual to see two fitting curves, so please explain why two different results are presented.

**Reply:**

There are two fitting curves included in black. The red line is just to guide the reader to the 1 to 1 correlation.

**Comment:**

Figure 7: The caption mentions only zonal mean zonal winds, but meridional winds are also shown in the figure. Please correct the caption to reflect this.

**Reply:**

Corrected.

**Comment:**

Line 191: What is the "4-hour mean"? Please explain this in the text before introducing it.

**Reply:**

All measurements within a 4-hour time-altitude-longitude bin are median averaged to mitigate statistical outliers.

**Comment:**

Figure 8: Please explain the scientific value this figure in the text.

**Reply:**

The figure actually shows that TIDI winds can capture tidal features on both hemispheres when all quality controls are considered. Unfortunately, very often there are large gaps posing challenges to obtain mean winds and tides directly.

**Comment:**

Figure 8: The caption states that the composite winds for January are shown, but the data were averaged over 60 days. This is confusing, as January only has around 30 days. Please explain what is meant by "60-day averaging" for the month of January.

**Reply:**

We apply a 60-day running window. January has 31-days and, thus, statically presents the largest subset. December and February contribute with 15-days.

**Comment:**

Lines 260-261: The TIE-GCM already includes an option to use TIDI wind climatology for lower atmospheric boundary conditions. Please clarify this in the text.

**Reply:**

This indeed a valuable remark. We will rephrase.

**References:**

**Comment:**

Wu, C., & Ridley, A. J. (2023). Comparison of TIDI Line of Sight Winds With ICON-MIGHTI Measurements. Journal of Geophysical Research: Space Physics, 128(2), e2022JA030910. https://doi.org/10.1029/2022JA030910

Niciejewski, R., Wu, Q., Skinner, W., Gell, D., Cooper, M., Marshall, A., et al. (2006). TIMED Doppler Interferometer on the Thermosphere Ionosphere Mesosphere Energetics and Dynamics satellite: Data product overview. Journal of Geophysical Research, 111(A11), A11S90. https://doi.org/10.1029/2005JA011513

Wu, Q., Ortland, D. A., Killeen, T. L., Roble, R. G., Hagan, M. E., Liu, H.-L., et al. (2008). Global distribution and interannual variations of mesospheric and lower thermospheric neutral wind diurnal tide: 1. Migrating tide. Journal of Geophysical Research: Space Physics, 113(A5), https://doi.org/10.1029/2007JA012542

**Reply:**

These papers are already cited.

**Comment:**

Dhadly, M., Jones, M., Emmert, J., Drob, D., Budzien, S., Zawdie, K., & McCormack, J. (2024). Short-Term to Inter-Annual Variability of the Non-Migrating Tide DE3 From MIGHTI, SABER, and TIDI: Potential Tropospheric Sources and Ionospheric Impacts. Journal of Geophysical Research: Space Physics, 129(8), e2024JA032849. https://doi.org/10.1029/2024JA032849

**Reply:**

Low-latitude tides are not part of the manuscript.

**Comment:**

Dhadly, M. S., Emmert, J. T., Drob, D. P., McCormack, J. P., & Niciejewski, R. J. (2018). Short-Term and Interannual Variations of Migrating Diurnal and Semidiurnal Tides in the Mesosphere and Lower Thermosphere. Journal of Geophysical Research: Space Physics, 123(8), 7106–7123. https://doi.org/10.1029/2018JA025748

**Reply:**

This paper is also about tides and outside the main narrative of the manuscript.

**Comment:**

Dhadly, M. S., Englert, C. R., Drob, D. P., Emmert, J. T., Niciejewski, R., & Zawdie, K. A. (2021). Comparison of ICON/MIGHTI and TIMED/TIDI Neutral Wind Measurements in the Lower Thermosphere. Journal of Geophysical Research: Space Physics, 126(12), e2021JA029904. https://doi.org/10.1029/2021JA0299043

**Reply:**

This is paper is indeed highly relevant for the submitted manuscript.

**Reply to reviewer 2**

**General Reply:**

We thank the reviewer for the assessment of our manuscript and the appreciation of its content. The manuscript focuses on comparing mean wind climatologies derived from TIDI with those obtained from local meteor radars. A statistical quality control procedure was implemented to ensure reliable comparisons. The findings show strong agreement between TIDI measurements and meteor radar winds, particularly for zonal winds, where TIDI captures most seasonal patterns accurately. However, the instrument shows less precision in capturing meridional winds.

**General Comment:**

This paper generates TIDI wind climatologies that are compared with meteor radar winds from several mid-latitude stations. The analyses are very careful and thorough, and have been carried out with an eye toward calibrating TIDI's zero wind line. The authors find very good agreement between radar winds and point TIDI measurements. Composite seasonal evolutions are presented for TIDI zonal mean winds. The overall zonal wind climatology looks quite reasonable, but the meridional zonal mean wind is likely compromised by tides. This paper presents an important and long-overdue assessment/validation of TIDI winds against simultaneous ground-based winds. It should be published, after the authors consider the specific comments below, and the edits to the manuscript.

**Comment:**

Line 18: "Zonal winds show..." Start new paragraph.

**Reply:**

Corrected.

**Comment:**

Line 25: Define migrating versus nonmigrating tides.

**Reply:**

We added this explicitly in the manuscript.

**Comment:**

Line 40: "The goal of this study..." Move to line 39, top of paragraph.

**Reply:**

Corrected.

**Comment:**

Line 68: "There exist...": New paragraph here.

**Reply:**

Corrected.

**Comment:**

Line 100: "That exceed 120 m/s..." How did you decide on that number?

**Reply:**

The value of of 120 m/s was motivated by the meteor radar climatologies. Typical 1 hour mean winds basically never exceeded this threshold. Therefore, this seemed to be a logical choice to be implemented for TIDI as well.

**Comment:**

Section 3.2: Why are you comparing the radar seasonal winds with TIDI zonal mean winds, instead of TIDI winds at the radar sites?

**Reply:**

We performed a comparison of the local observations around the meteor radar locations in Figure 2 and 3. However, even when using the 60-day averaging window, there would be a 2-hour gap around the morning and evening terminator due to the TIDI issues at these times. Therefore, we decided that a zonal mean could compensate for these data gaps and provide a more robust average for the mean winds rather than the local and confined bins to some longitudes and latitudes.

**Comment:**

Lines 129-136: Not clear what is meant by nonmigrating tides modeled as "noise".

**Reply:**

We noticed artificial jumps in the wind time series associated with certain local times. These jumps overlayed a potential tidal signal and prevented us from analyzing and comparing specific tidal modes. However, this procedure mitigated the biases introduced by the instrument and resulted in more reliable zonal mean winds. The meridional winds have much smaller magnitudes and, thus, the correlation is decreased. Furthermore, atmospheric wave effects due to the sampling might still be more problematic. This needs to be investigated in a separate study.

I will add it in the manuscript.

**Comment:**

Since you are showing zonally averaged TIDI winds in Figures 4-6, from which nonmigrating tides are filtered, I suggest removing verbiage about nonmigrating tides (seen in the pdf attachment with the edits).

**Reply:**

Corrected.

**Comment:**

Figures 4 and 7: TIDI zonally averaged meridional winds show strong signs of aliasing by the migrating diurnal tide. In particular, the zero values at the equator, the latitudinal asymmetry, and the wind reversals with altitude near 90 km. I suggest testing the analysis and quantifying this possible aliasing effect by sampling WACCM-X meridional winds identically to TIDI.

**Reply:**

Thank you for the observation and the suggestion. This needs to be investigated in a separate study.

**Comment:**

Line 232: "...mixing between migrating and non-migrating tides..." Can you explain how these two categories of waves "mix", even with short term variability? Migrating tides alias to the zonal mean, whereas nonmigrating tides alias to integer wavenumbers, which are quite distinct from the longitudinal mean. It seems to me that "mixing" would occur only if TIDI is not sampling around a full longitude circle. Are there situations when this happens, for example, near the South Atlantic Anomaly?

**Reply:**

The term mixing refers to the incomplete longitudinal sampling of TIDI. Due to the filter criteria that we applied there are sometimes gaps for certain longitudes around the terminator. These data gaps are posing problems to the tidal fits. Furthermore, the analysis of migrating and non-migrating tides assumes phase stability of all modes within the sampling window. However, this assumption is certainly not holding for each meteorlogical situation, where rapid changes in the background wind induce changes in the phases of different tidal modes giving rise to aliasing effects as the phase changes could also be seen as a change in the wavenumber.

See both articles:

https://acp.copernicus.org/articles/20/11979/2020/acp-20-11979-2020.html

https://agupubs.onlinelibrary.wiley.com/doi/full/10.1029/2023JA031680

**Reply to topic editor**

**General comment:**

Currently your manuscript requires major revision. Please send us a revised manuscript with the changes highlighted.

I would like to request you to reflect your opinion/explanation on TIDI winds bias issues and possible measurement errors around the solar terminator in the discussion section so that the pitfalls of this issue are also clarified to the readers.

Your modifications will be shared with one of the reviewers and myself.

**General response:**

This point is added in the discussion of the new version in the sentence:

Apart from the bias in TIDI coldside vector winds during forward flight, another possible bias might originate from wind measurements near terminators where there is a gradient in emissions. One possible solution is to exclude data where the solar beta angle is high, but this data is not available to us.

---

## Author Response (AR2)

**Reply to reviewer 1**

**General reply:**

We appreciate the reviewers time for reading the manuscript and providing feedback to our analysis. As mentioned in a previous reply, modeling the issues of the hot-cold zero-line calibration for certain orbits as tidal mode and applying the ASF to remove this bias to obtain reliable mean winds is equivalent to the solar beta angle filtering. So, there is no contradiction at all. However, to avoid further discussion, we repeated the entire analysis using only a solar beta angle filter as suggested in Dhadly et al., 2021 and compared the results to our filtering. To keep things consistent the climatological fits both, include the ASF. Given the agreement, we are confident that both methods are applicable as a quality control mechanism for TIDI winds.

**General Comment:**

I appreciate the authors taking the time to respond to my comments, suggestions, and concerns. However, my key concerns remain, and I am not fully satisfied with the responses to the major issues raised in the previous revision. The authors cited the lack of orbital characteristics in the data files as a limitation for making improvements. However, I have checked that all necessary data to address bias (including solar beta to determine the terminator locations) is available in TIDI wind data files located at the data link provided in the manuscript. Additionally, the authors emphasized that the previous cited studies are focused on a latitude band of approximately 10S to 40N, whereas their study examines middle to high latitudes. However, this reasoning is not valid, as the bias associated with TIDI's zero-wind problem on one of the telescopes is independent of location and persists regardless of the satellite's location on the globe. Furthermore, as previously suggested, July is possibly not an ideal month for TIDI comparisons, as TIDI may be observing near the terminator by the end of the month. The authors have not addressed this concern in their responses.

**Reply:**

We have been mistaken looking at the data base and searched for the solar beta angle in the group data of the netcdf file instead of the header attributes.

Comparison of statistical method Gauthier et al. versa solar beta angle Dhadly et al.:

First, we computed the climatological database applying only the filter as described in the submitted manuscript and a second analysis only using the solar beta angles (<55°). We show a representative correlation for the Collm location.

Dhadly et al., 2021 - Solar beta angle <55°

[Figure]

Gauthier et al. statistical filter + ASF

[Figure]

We also compared the latitudinal climatologies leveraging both methods, which resulted in nearly identical structures. However, due to the lower statistical limit of the minimum required number of measurements the solar beta angle filter resulted in an increased coverage during the hemispheric winter condition.

[Figure]

**Reply to reviewer 2**

**General Reply:**

We thank the reviewer for their assessment and appreciation of our manuscript. It compares mean wind climatologies from TIDI's extensive dataset and local meteor radars, with a statistical quality control procedure. The results show strong agreement, especially for zonal winds, where TIDI captures most seasonal patterns, though it is less precise for meridional winds.

**General Comment:**

The authors present a very nice comparison of TIDI data to meteor radar data for the lifetime of the TIDI instrument. They examine the global climatology derived from measurements. Zonal winds are found to have better agreement than meridional and some expected climatology results are seen. I recommend this paper for publishing with the very minor corrections suggested below, which are mostly proof-reading.

**Comment:**

Line 74: it is unclear what "This" is. The NCAR method?.

**Reply:**

We have revised this paragraph to be more clear which method refers to which organization.

**Comment:**

Figure 2, 3: Are a) and d) supposed to be the same plot? Maybe just have it once?

**Reply:**

We remove panel d) from the panel to avoid duplication of information.

**Comment:**

Line 128: Spacing issues with parentheses. Is "released" the word you want?

**Reply:**

We rephrased this sentence and corrected the parentheses.

**Comment:**

Line 160: Missing an "and", perhaps?

**Reply:**

Corrected.

**Comment:**

Line 180: "Figures 5 and Figure 6". Figures 5 and 6? Figures 5 and Figures 6? Plus, there's an extra space I think before 6.

**Reply:**

Done.

**Comment:**

Discussion: Are there suggestions/conjectures as to why the meridional winds are less correlated that could be stated in the discussion of those winds?

**Reply:**

Meridional winds show lower correlation than meteor radar due to their weaker magnitudes on seasonal time scales, compared to zonal winds. This makes them more susceptible to local atmospheric wave variability and sampling effects.

We added it in the discussion part of the manuscript.